# Connective differences between patients with depression with and without ASD: A case-control study

Tomoki Kaneko[1]*, Toshinori Nakamura[2], Akiko Ryokawa[2], Shinsuke Washizuka[2], Yoshihiro Kitoh[3], Yasunari Fujinaga[1]

1 Department of Radiology, Shinshu University School of Medicine, Matsumoto, Nagano, Japan,
2 Department of Psychology, Shinshu University School of Medicine, Matsumoto, Nagano, Japan,
3 Department of Radiology, Shinshu University Hospital, Matsumoto, Japan

* tk55@shinshu-u.ac.jp

## Abstract

### Background

Researchers find it difficult to distinguish between depression with ASD (Depress-wASD) and without ASD (Depression) in adult patients. We aimed to clarify the differences in brain connectivity between patients with depression with ASD and without ASD.

### Methods

From April 2017 to February 2019, 22 patients with suspected depression were admitted to the hospital for diagnosis or follow-up and met the inclusion criteria. The diagnosis was determined according to the Diagnostic and Statistical Manual of Mental Disorders-5 by skilled psychiatrists. The Hamilton Depression Rating Scale (HAM-D), Young Mania Raging Scale (YMRS), Mini-International Neuropsychiatric Interview, Parent-interview ASD Rating Scale-Text Revision (PARS-TR), and Autism-Spectrum Quotient-Japanese version (AQ-J) were used to assess the patients' background and help with diagnosis. Resting-state functional magnetic resonance imaging (rs-fMRI) was performed using the 3-T-MRI system. rs-fMRI was processed using the CONN functional connectivity toolbox. Voxel-based morphometry was performed using structural images.

### Results

No significant difference was observed between the Depress-wASD and Depression groups using the HAM-D, YMRS, AQ-J, Intelligence Quotient (IQ), and verbal IQ results. rs-fMRI for the Depress-wASD group indicated a positive connection between the salience network (SN) and right supramarginal gyrus (SMG) and a negative connection between the SN and hippocampus and para-hippocampus than that for the Depression group. No significant structural differences were observed between the groups.

**Data Availability Statement:** Data cannot be shared publicly because of not getting consent about the data publicity. In addition, when we obtained consent, the consent form did not contain

the clause for public sharing. After consultation with the Ethics Committee of Shinshu University Hospital, the data are available to researchers who meet the criteria for access to confidential data from the responsible party listed below. The image data for this study are stored on the data server of the Division of Radiology, Shinshu University Hospital. The person responsible for managing the image data used in this study is Yasuo Adachi. He is a radiological technologist at the section chief of the MRI in the division of radiology in the shinshu-university hospital. The e-mail address for contacting Yasuo Adachi is yadachi@shinshu-u.ac.jp.

**Funding:** The authors received no specific funding for this work.

**Competing interests:** The authors have declared that no competing interests exist.

**Abbreviations:** ASD, autism spectrum disorder; AQ-J, Autism-Spectrum Quotient-Japanese version; CI, confidence interval; CSF, cerebrospinal fluid; DMN, default mode network; DSM-5, Diagnostic and Statistical Manual of Mental Disorders-5; FDR, false discovery rate; HAM-D, Hamilton Depression Rating Scale; GM, gray matter; M.I.N.I, Mini-International Neuropsychiatric Interview; PARS, Parent-interview ASD Rating Scale-Text Revision; PFC, prefrontal cortex; ROI, region of interest; SMG, supramarginal gyrus; SN, salience network; TE, echo time; TI, inversion time; TR, repetition time; ts-fMRI, task-based functional magnetic resonance imaging; WM, white matter; YMRS, Young Mania Raging Scale.

## Conclusions

We identified differences in the SN involving the SMG and hippocampal regions between the Depress-wASD and Depression groups.

## Introduction

According to the Diagnostic and Statistical Manual of Mental Disorders-5 (DSM-5), autism spectrum disorder (ASD) is classified as a neurodevelopmental disorder. It is defined as "an impairment in social communication" and "limited interest," and these impairments appear in the first 2 years of life [1]. In a survey of 3,954 of 5,016 5-year-old children, Sato et al. reported on a crude ASD prevalence of 1.73% (95% confidence interval [CI] 1.37–2.10%) and a male-female ratio of 2.22: 1. The prevalence after statistically adjusting for the non-participating children was estimated to be 3.22% (95% CI 2.66–3.76%) [2].

Approximately 30% of young adults with ASD have co-occurring psychiatric disorders [3]. ASD affects intellectual ability in approximately 30% of the cases and poses a high risk of depression. Depression is observed in 19.8% of the patients with ASD, compared with that in only 6.0% of the healthy controls [4]. The risk of a depression diagnosis is reportedly higher in patients with ASD without intellectual disability than in those with ASD and intellectual disability [4]. In addition, no effective treatment exists for this condition, and these impairments supposedly continue throughout a patient's lifetime. Thus, optimized life support is likely to exert a positive effect on the patients' quality of life [5].

Patients with ASD have reported a high rate of depression in studies on ASD complications [6, 7]. A meta-analysis on anxiety and depression in adults with ASD reported 23% and 37% pooled estimation of current and lifetime prevalence of depressive disorder, respectively [8]. In comparison to pediatric patients with ASD who develop depression, some adult patients with ASD may be diagnosed with depression and later indicate ASD [9]. Adult ASD with depression is difficult to diagnose because a physician has limited specific information regarding a patient's childhood [10]. The DSM-5, International Statistical Classification of Disease and Related Health Problems, Tenth Revision, and DSM-IV-TR are used to diagnose ASD in adults; nonetheless, they have different sensitivity and specificity [11]. The Autism-Spectrum Quotient (AQ), Parent-Interview ASD Rating Scale-Text Revision (PARS), and Wechsler Intelligence Test are screening instruments used to assess ASD in Japan [12, 13]. Even if the AQ score of the patient exceeds the cut-off ($\geq$33 points), further evaluation is required by a skilled physician for ASD diagnosis. The PARS is an excellent indicator of sensitivity and specificity in ASD diagnosis in adults because it provides information on the natural developmental characteristics of children [13]. However, it requires a direct interview with the primary caregiver, which is difficult in the current situation with limited mobility and visitations.

Task-based functional magnetic resonance imaging (ts-fMRI) is used to assess morphological and functional abnormalities, comparing the characteristics of patients with ASD and those of healthy individuals [14–16]. ASD has numerous comorbidities, and some cases are treated as depression, particularly when depression is in the foreground. This is because it is difficult to diagnose the underlying ASD. Resting-state fMRI (rs-fMRI) is a technique for recording slow-frequency fluctuations of brain activity and analyzing the connectivity of brain regions [17]. We aimed to clarify differences between networks in patients with depression with ASD (Depress-wASD) and those without ASD (Depression).

## Materials and methods

### Patients

We recruited 25 consecutive individuals who were suspected of having depression and who visited the Department of Psychiatry at Shinshu University Hospital (Nagano, Japan) for a diagnosis or follow-up between April 2017 and February 2019. Of them, 22 individuals met the following criteria:

1. Age ≥20 years while providing consent

2. Planned to be treated at our hospital

3. Received sufficient explanation about this study and provided informed consent. If consent could not be obtained directly because of their medical condition (e.g., the inability to speak or write), it was obtained from a surrogate.

The patient's medical condition was investigated by skilled psychiatrists. They determined the presence of any metal in the body that would contraindicate an MRI examination. They confirmed the patient's free will to participate in the study and whether the patient could keep calm during the examination. Informed consent was obtained from each participant. We obtained consent for publication from each patient. All procedures were performed in accordance with relevant guidelines/regulations of the ethics committee of Shinshu University School of Medicine and the tenets of the Declaration of Helsinki.

We excluded three patients for the following reasons: incomplete data on fMRI (n = 1), incomplete psychological test result (n = 1), and left handedness (n = 1).

### Diagnostic criteria and psychological examinations

Diagnosis was determined according to the DSM-5 by assessing the medical history and continuous observation of psychiatric symptoms by skilled psychiatrists. Because there were no physical criteria to diagnose these neurodevelopmental disorders, skilled psychiatrists used the following clinical information: depression severity rating according to the Hamilton Depression Rating Scale (HAM-D), manic severity rating according to the Young Mania Raging Scale (YMRS), The Mini-International Neuropsychiatric Interview (M. I. N. I.), Parent-interview ASD Rating Scale-Text Revision (PARS-TR), and AQ-Japanese version (AQ-J). In addition, clinical tests necessary for diagnosis and treatment were performed as needed. Finally, 8 out of 22 patients were diagnosed with Depress-wASD by the psychiatrists.

### Neuroimaging acquisition

We used a 3-Tesla MRI system (Prisma system; Siemens, Erlangen, Germany) with a 64-channel head coil. Anatomical images were obtained using a three-dimensional T1-weighted Magnetization-Prepared Rapid-Gradient Echo sequence (repetition time [TR] = 1,900 ms, echo time [TE] = 2.9 ms, inversion time [TI] = 960 ms, and flip angle = 9°, $1 \times 1 \times 1$-mm resolution). The acquisition time was 5 min 50 s. We performed rs-fMRI using a two-dimensional gradient echo-planar sequence (TR = 2,500 ms, TE = 30 ms, and flip angle = 80°), voxel size = $3.3 \times 3.3 \times 3.2$ mm, and acquisition time = 10 min. The patients were instructed to remain awake and to look at one point. We adopted an auto-discarding system that scanned the first 10 volumes and discarded them to allow for magnetic field stabilization.

### Resting-state functional MRI analysis

We used the CONN functional connectivity toolbox 18a (www.nitrc.org/projects/conn) in the MATLAB R2018a environment (Marth Works, USA).

## Preprocessing

Functional and structural images were preprocessed using the default preprocessing CONN pipeline [18]. This preprocessing enabled us to rectify each patient's blood oxygen level-dependent (BOLD) signal to the template coordinates (Figs 1 and 2).

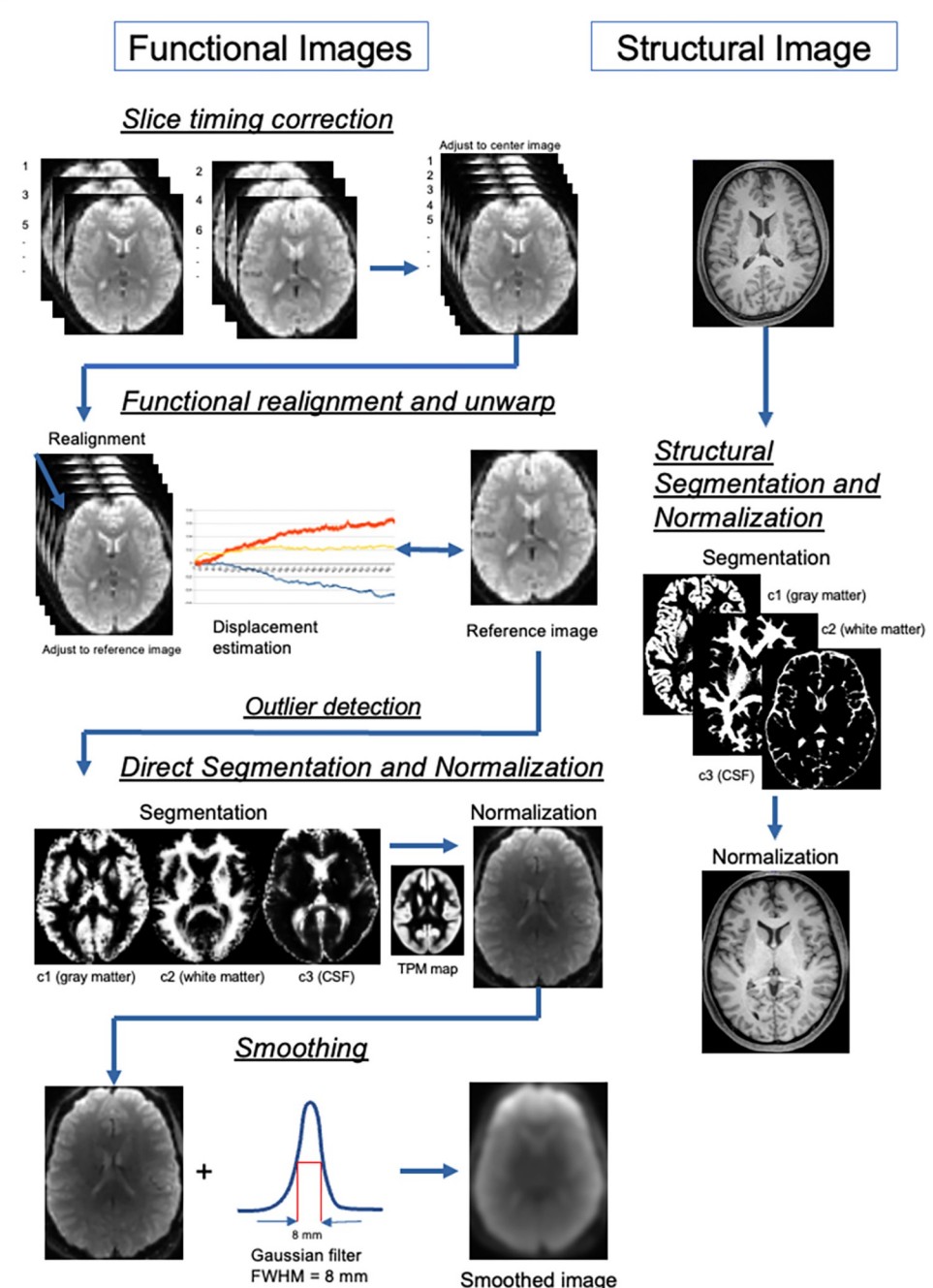

**Fig 1. The preprocessing flow chart.** Functional and structural images are normalized and segmented directory to the Montreal Neurological Institute (MNI) space. Before normalization to the MNI space, the functional images are corrected for slice timing and realigned to the reference, and outlier components are detected.

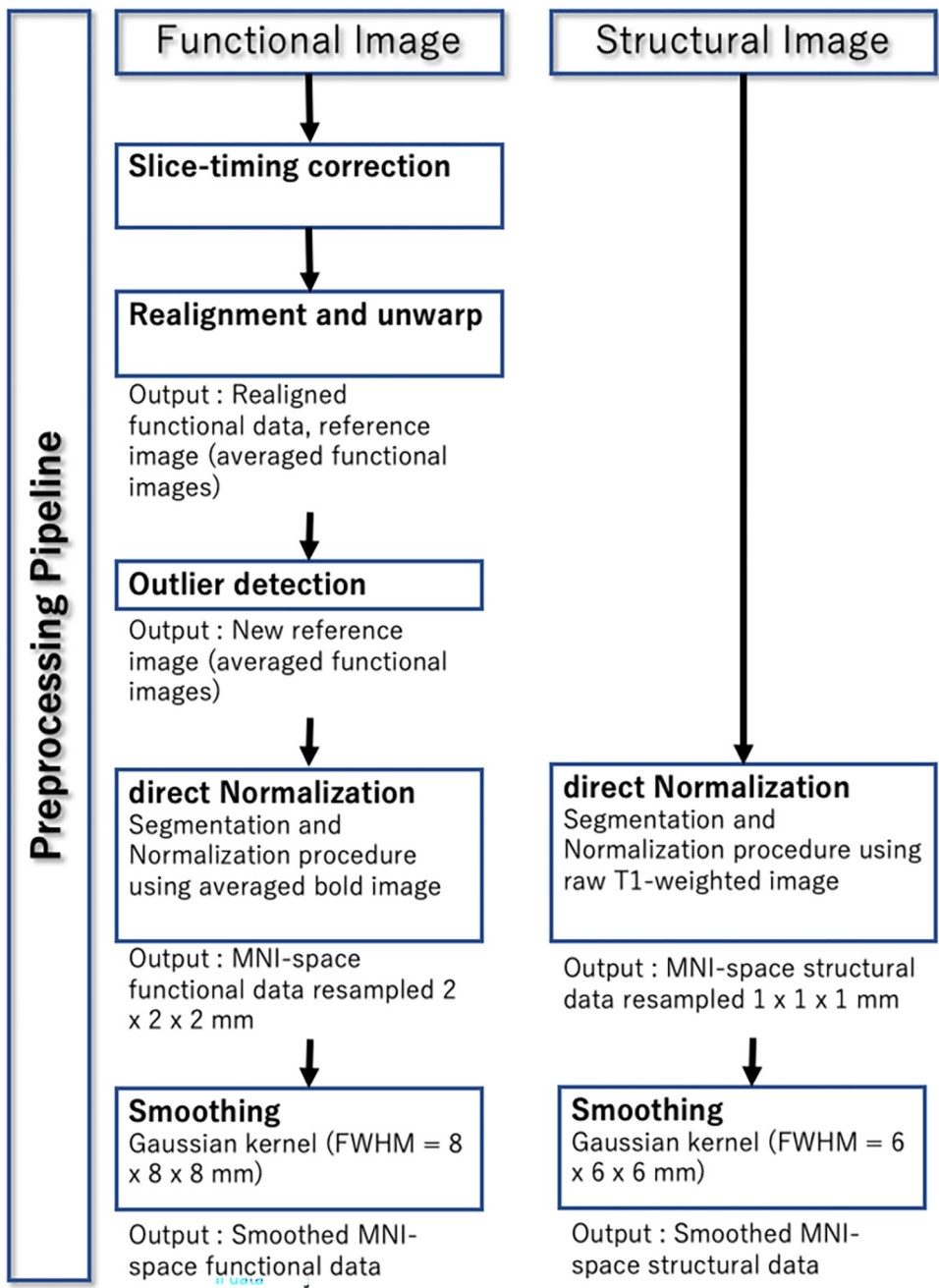

**Fig 2. The preprocessing flow chart detailing the image information.** The block images depict the process and major output images for later statistical analyses.

1. Functional direct preprocessing pipeline: First, we executed "slice timing correction." A functional head volume image was obtained through inter-leaved acquisition. Subsequently, we fixed the slice timing, paying attention to prevent mixing of the multiple timing points with the following process. In the subsequent realignment and unwarping step, we realigned and unwrapped the functional images, estimating and correcting for subject motion. The centering step roughly corrected the functional images along the anterior-

posterior commissure line. During outlier detection, potential outlier scans of framewise displacements >0.9 mm or BOLD signal changes >5 s.d. were detected and flagged as potential outliers. Throughout the process, functional images were normalized into the standard MNI space using SPM12 (Statistical Parametric Mapping) and were segmented into gray matter (GM), white matter (WM), and cerebral spinal fluid (CSF). Direct processing ensured that these images were normalized and segmented separately without any relation to structural preprocessing. Finally, they were smoothed with an 8-mm Gaussian kernel full with half maximum (FWHM) to increase the signal-to-noise (S/N) ratio.

2. Furthermore, the structural images were normalized and segmented as functional processing using SPM unified segmentation and normalization procedures.

## Denoising

To minimize the influence of artifactual factors on functional connectivity measures, we adopted a default denoising pipeline combining the two steps, namely, linear regression and temporal band-pass filtering.

1. Linear regression involved the following steps: anatomical component-based noise correction procedure (aCompCor), noise components from the cerebral WM and cerebrospinal areas, estimating patient-motion parameters, identifying outlier scans or scrubbing, and first-order linear session effects.

2. Temporal band-pass filtering and temporal frequencies <0.008 Hz or >0.09 Hz were removed from the BOLD signal to focus on slow-frequency fluctuations.

## First- and second-level analysis

1. First-level analysis: Region of interest (ROI)-to-ROI connectivity (RRC) matrices were estimated by characterizing the functional connectivity between each pair of regions among 164 HPC-ICA networks and Harvard-Oxford atlas ROIs [18, 19]. These RRC matrices represented the connectivity level of each pair of ROIs, and each element in a matrix was defined as the Fisher-transformed bivariate correlation coefficient between a pair of ROI BOLD time series [20].

2. Second-level analysis: We compared the pairwise ROI-to-ROI connectivity strength values across the diagnostic groups (Depression with ASD vs. Depression without ASD (Depression)) with adjustment for the age, sex, YMRS, IQ, verbal IQ, AQ-J, HAM-D, and PARS scores as the covariates. Second-level analyses were performed using a General Linear Model (GLM) [20]. Connection-level hypotheses were evaluated using multivariate parametric statistics with random-effects across participants and sample covariance estimation across multiple measurements. We drew inferences at the level of individual clusters (groups of similar connections). Cluster-level inferences were based on parametric statistics within and between each pair of networks (Functional Network Connectivity), with the networks identified using a complete-linkage hierarchical clustering procedure [21, 22] based on the ROI-to-ROI anatomical proximity and functional similarity metrics [20]. The results were thresholded using a combination of a p-value <0.05 connection-level threshold and a familywise corrected p-FDR <0.05 cluster-level threshold [23] (Fig 3).

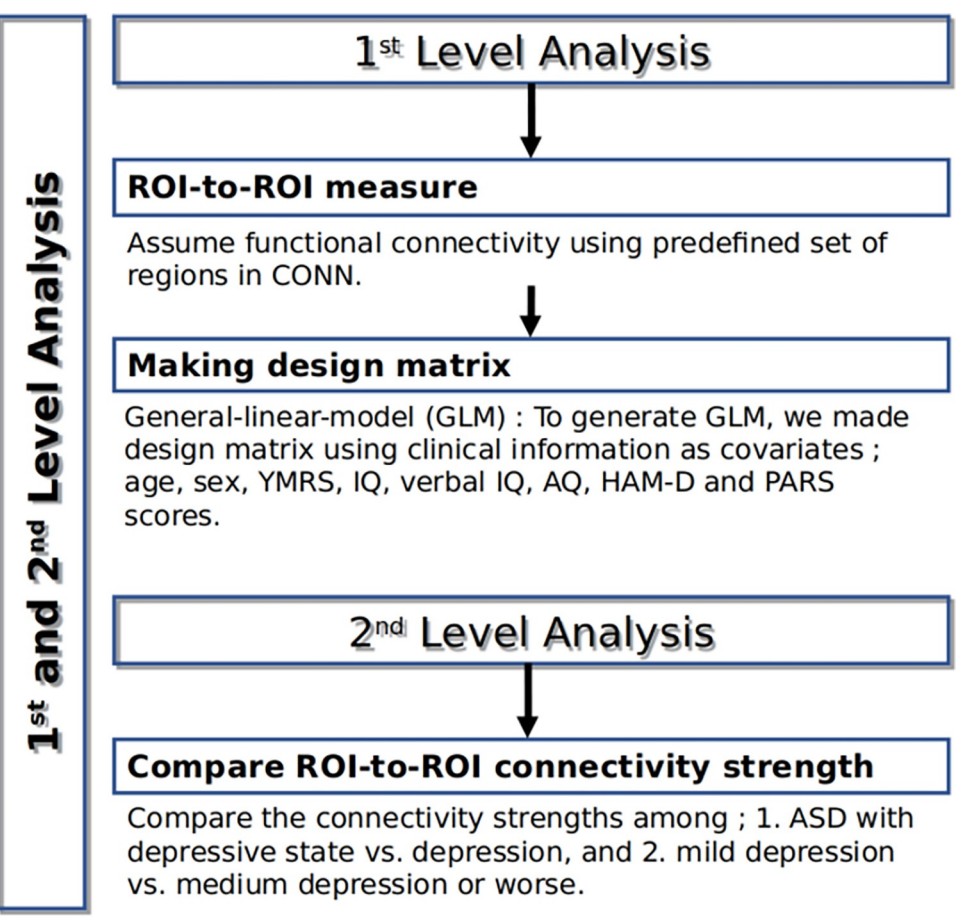

**Fig 3. The preprocessing flow chart.**

## Structural analysis

**Voxel-based morphometry.**   We used SPM12 to estimate the differences between Depress-wASD and Depression. Preprocessed structural images were smoothed with a 6-mm Gaussian kernel of FWHM to increase the S/N ratio.

Post-processed structural images were divided into two groups as follows: Depress-wASD and Depression. First, we constructed a design matrix to adopt the GLM [20] and to identify regions significantly related to the differences between the groups. Furthermore, we performed an estimation to validate the independently distributed residuals. Finally, we performed the two-sample t-test using the standard parametric procedure to test the hypothesis [24].

This study was approved by the ethics committee of Shinshu University School of Medicine (3612). Written informed consent was obtained from all participants or their parents.

## Results

### Demographics

No significant difference was observed between the Depress-wASD and Depression groups in terms of the sex, age, HAM-D, YMRS, AQ, PARS, IQ, and verbal IQ (Table 1). All participants remained calm during the examination. No data were excluded because of severe motion artifacts or other artifacts.

Table 1. Demographics.

|  | Non-ASD | ASD | p-value |
|---|---|---|---|
| n | 14 | 8 |  |
| Male:female | 6:8 | 5:3 | 0.66[1] |
| Age median [min, max] | 39 [22, 52] | 25.5 [21, 41] | 0,06[2] |
| HAM-D median [min, max] | 17 [8, 28] | 16.5 [9, 29] | 0.89[2] |
| YMRS median [min, max] | 0 [0, 5] | 0 [0, 6] | 0.68[2] |
| AQ median [min, max] | 25 [10, 40] | 27 [15, 39] | 0.91[2] |
| PARS median [min, max] | 6.5 [3, 23] | 12.5 [6, 33] | 0.13[2] |
| IQ median [min, max] | 96 [76, 114] | 97 [54, 112] | 0.85[2] |
| Verbal IQ median [min, max] | 98.5 [75, 207] | 101.1 [63, 104] | 0.48[2] |

HAM-D, Hamilton Depression Rating Scale; YMRS, Young Mania Rating Scale; AQ, Autism-Spectrum Quotient; PARS, Parent-interview ASD Rating Scale; and IQ, Intelligence Quotient. 1) Fisher's exact test; 2) Mann–Whitney U test

## Depress-wASD versus depression

In this analysis, the left salience network (SN) displayed increased connectivity to the right supramarginal gyrus (SMG) (p-FDR<0.04*) and decreased connectivity to the left hippocampus (p-FDR<0.02*) and para-hippocampus (p-FDR<0.02*). However, no significant increase or decrease was observed in the activity of each seed (Fig 4, Table 2).

## Volumetric findings

We observed no significant difference between the Depress-wASD and Depression groups.

## Discussion

We performed an ROI-to-ROI analysis to investigate the difference in connectivity between Depress-wASD and Depression groups. We hypothesized that the DMN of the Depression group, termed as characteristic connectivity, would demonstrate higher connectivity than that of the Depress-wASD group; nonetheless, no significant difference was observed around the DMN. In contrast, the SN demonstrated increased connectivity to the right SMG and decreased connectivity to the left hippocampus and para-hippocampus.

In this study, the numbers of men and women were equal in both the groups. We identified relatively more men with ASD and relatively fewer women with depression; however, there was no statistically significant difference in sex distribution. Generally, ASD is >4 times more common among men than among women, and depression is twice as common among women [25–27]. The equal number of men and women in this study was in line to that in a previous study because of the opposite sex distribution between ASD and depression [25–27]. However, our sample size was relatively small to analyze sex differences. Researchers should consider other psychiatric disorders upon identifying no sex differences in the population presenting with depression.

## Depress-wASD versus depression

The Depress-wASD group displayed increased connectivity between the SN and right SMG and decreased connectivity between the SN and hippocampal regions. No significant difference was observed in the verbal IQ between the groups.

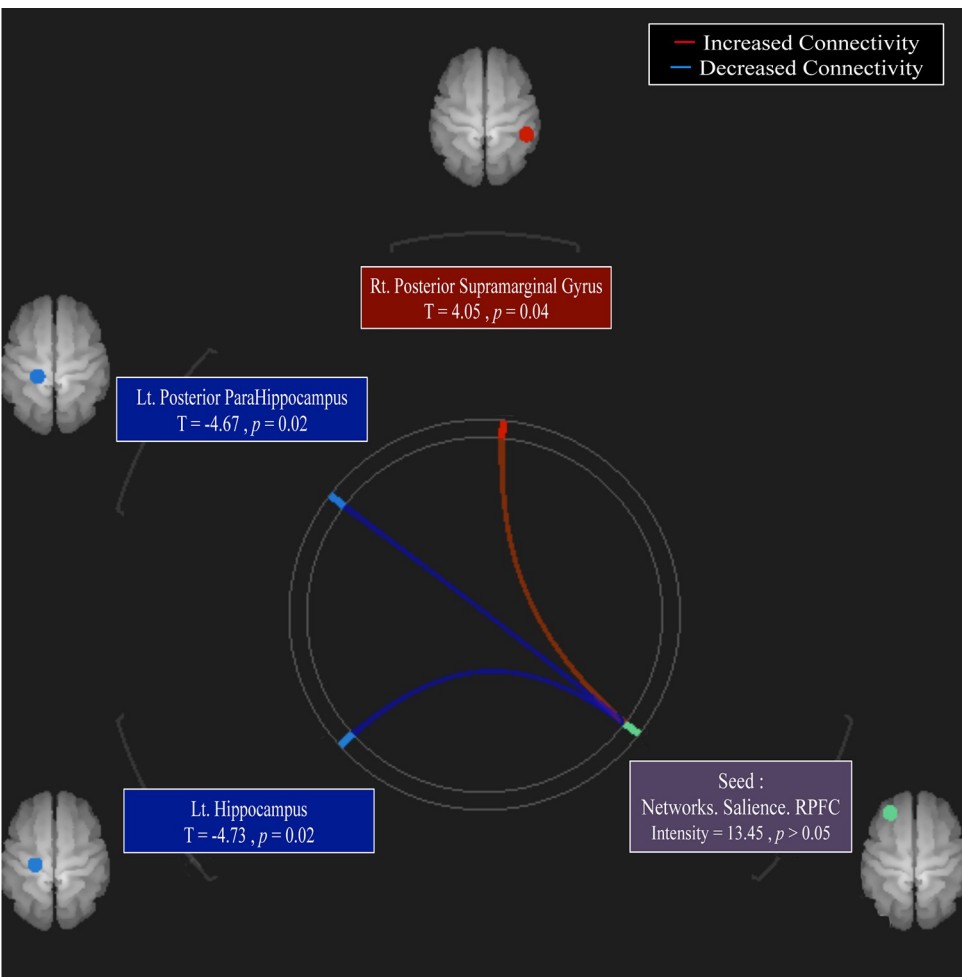

**Fig 4. Increased connectivity in the red lines and decreased connectivity in the blue lines.** The purple block depicts the seed point with statistics of the strength of the seed activity. Each colored block indicates the increased and decreased strength of connectivity with statistics (T- and *p*-values), respectively. Increased connection is observed between the salience network (SN) and the right posterior SMG. Decreased connection is observed between the SN and the left hippocampus and para-hippocampus gyrus. There is no significant increase/decrease in each seed.

**Table 2. Functional connectivity between patients with depression with and without ASD.**

|  | Statistic | p-FDR |
|---|---|---|
| **Seed Networks. Salience. RPFC** | **F = 5.49**<br>**Intensity = 13.45**<br>**Size = 3** | - |
| • Hippocampus (left) | T = -4.73 | 0.02 |
| • Posterior para-hippocampus (left) | T = -4.67 | 0.02 |
| • Posterior supramarginal gyrus (right) | T = 4.05 | 0.04 |

FDR, false discovery rate, RPFC, rostral prefrontal cortex. The statistical results can be thresholded using a combination of connection-level threshold; F-test on the network-based statistics.

The SN refers to a series of brain networks comprising the anterior cingulate cortex, anterior insula, and orbitofrontal cortex; it contributes to numerous complex brain functions, including communication, social behavior, and self-awareness by integrating sensory, emotional, and cognitive information [28, 29]. rs-fMRI for 17 children with ASD displayed greater connectivity between the SN regions (SMG to rostral prefrontal cortex [rPFC]) and lower verbal IQ than those in control patients [30]. In contrast, in typically developing patients, the functional connectivity between the SN and medial PFC decreased with age; however, no significant change was observed in the patients with ASD [31], consistent with our findings. Within the SN, the rPFC supports a cognitive system [32, 33]. The medial area of the PFC supports processes related to stimulus-oriented attending, i.e., the behavior required to concentrate on current sensory input. By contrast, stimulus-independent attending is the mental processing that accompanies self-generated or self-maintained thoughts [32]. Moreover, multitasking is significantly impaired in patients with rPFC lesions, despite no significant impairment in remembering the task rules [34]. Reports of high connectivity in the SN-rSMG in children with ASD indicate residual childhood connectivity or delayed development.

We discuss the high connectivity between the SN and the right SMG and the low connectivity between the SN and memory areas from brain function development during language acquisition in the Japanese. Japanese infants aged from 4 to 5 years can segment words using voice input by mora units [35]. When they begin to learn the letters, they connect a word with phonology. In the initial stage of character acquisition, each character is converted phonologically and read sequentially; however, once the reading becomes proficient, meaning processing is possible only with character form information of the word without undergoing phonological conversion [36]. In an fMRI study comprising 37 right-handed adults, 95% of them demonstrated dominant speech and language function in the left hemisphere [37]. Hartwigen et al. mentioned that both the left and right SMGs were required for phonological decisions [38]. Our present results suggested that the network connectivity persists in adult patients with depression and ASD, and this persistence disturbs the functional differentiation that should be obtained by the functional localization observed in healthy adults. Moreover, this network demonstrated decreased connectivity to the hippocampus, which may lead to the disruption of phonological decisions. Our patients demonstrated normal verbal IQ; however, those with ASD may have developed depression because they managed to adapt to the environment, despite a disrupted phonological network.

Reports on ASD in children indicate that depression assessment is difficult because of nonspecific symptoms and overlapping phenotypic systems in ASD [39]. In a population-based birth cohort study, Maja et al. reported that approximately half of the participants had normal or high IQ, which increases the risk of not being diagnosed with ASD [40].

ASD is a spectrum of disorders, and our results capture only one aspect. However, changes in the functional brain connectivity may be a risk factor for the development of depression even if the patients appear to be well adjusted. Screening at an early stage may be necessary to enhance the effectiveness of therapeutic interventions.

We performed VBM using SPM12 to compare the structural differences between the Depress-wASD and Depression groups. Our results demonstrated no significant structural differences between these groups, despite previous reports on variable structural abnormality in patients with ASD presenting with increased or decreased volumes in the specific areas of the brain [41–44]. In previous studies, patients with ASD had varied age and clinical background (comorbidity, educational, and genetic profiles). Ecker et al. mentioned that patients with ASD had increased GM volume in the anterior temporal and dorsolateral prefrontal regions and decreased volume in the occipital and medial parietal regions, compared with healthy controls, despite no significant difference in the whole brain volume [41]. In the present study, patients

with ASD demonstrated no structural differences from those with depression. However, patients with depression display significant volume reductions than healthy controls, particularly in the hippocampus and amygdala [45, 46]. The differences in morphological alterations in each study could be attributed to variances in the clinical background. Therefore, our results indicated some structural abnormality in patients with ASD that should be reexamined in healthy controls.

## MRI of Depress-wASD

rs-fMRI requires a longer scanning time to obtain more information and achieve highly reproducible connectivity, compared with other routine sequences. Therefore, patients are instructed to remain awake and keep calm during the examination [47]. In this study, the patients did not present with ASD symptoms and were suspected of having depression. Despite including the patients with low IQs, they understood the explanation on the MRI procedure and were able to keep calm during the examination. Furthermore, we considered the age of the patients to account for the lack of severe movements or complaints. Our patients only displayed a small proportion of the characteristics of adult-onset ASD.

## Limitations

The limitations of this consecutive case series study include a small sample size. Accordingly, comparison between the ASD groups and the age-matched control group with a larger sample size is needed to confirm our findings. We intend to investigate the phonological function regarding language processing between the Depress-wASD group and age-matched normal participants using task-based fMRI.

## Conclusions

The Depress-wASD and Depression groups demonstrated differences in the SN involving the SMG and hippocampal regions. Our findings suggest that an immature phonological network could be one of the potential causes of depression in adult ASD.

## Supporting information

**S1 File. The signal-to-noise ratio of the each preprocessing process.**
(PDF)

## Author Contributions

**Conceptualization:** Toshinori Nakamura, Akiko Ryokawa, Shinsuke Washizuka.

**Data curation:** Tomoki Kaneko, Akiko Ryokawa, Yoshihiro Kitoh.

**Formal analysis:** Toshinori Nakamura.

**Investigation:** Tomoki Kaneko, Toshinori Nakamura, Akiko Ryokawa.

**Methodology:** Tomoki Kaneko, Toshinori Nakamura, Akiko Ryokawa.

**Project administration:** Tomoki Kaneko.

**Software:** Tomoki Kaneko.

**Supervision:** Toshinori Nakamura, Shinsuke Washizuka, Yasunari Fujinaga.

**Validation:** Tomoki Kaneko, Toshinori Nakamura, Yasunari Fujinaga.

**Writing – original draft:** Tomoki Kaneko.

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
