## [Decision Letter · Decision Letter 0]

23 Sep 2022

PONE-D-22-14224Connective differences between autism spectrum disorder with depressive state and depression: case-control study.PLOS ONE

Dear Dr. Kaneko,

Thank you for submitting your manuscript to PLOS ONE. After careful consideration, we feel that it has merit but does not fully meet PLOS ONE’s publication criteria as it currently stands. Therefore, we invite you to submit a revised version of the manuscript that addresses the points raised during the review process.

We look forward to receiving your revised manuscript.

Kind regards,

Jerritta Selvaraj

Academic Editor

PLOS ONE

Journal Requirements:

2. Please describe in your methods section how capacity to provide consent was determined for the participants in this study. Please also state whether your ethics committee or IRB approved this consent procedure. If you did not assess capacity to consent please briefly outline why this was not necessary in this case.

Please also note that PLOS ONE has specific guidelines on code sharing for submissions in which author-generated code underpins the findings in the manuscript. In these cases, all author-generated code must be made available without restrictions upon publication of the work. Please review our guidelines at https://journals.plos.org/plosone/s/materials-and-software-sharing#loc-sharing-code and ensure that your code is shared in a way that follows best practice and facilitates reproducibility and reuse.

"No financial disclosure."

Reviewers' comments:

Reviewer's Responses to Questions

**Comments to the Author**

1. Is the manuscript technically sound, and do the data support the conclusions?

Reviewer #1: Yes

Reviewer #2: Partly

2. Has the statistical analysis been performed appropriately and rigorously? 

Reviewer #1: I Don't Know

Reviewer #2: Yes

3. Have the authors made all data underlying the findings in their manuscript fully available?

Reviewer #1: No

Reviewer #2: No

4. Is the manuscript presented in an intelligible fashion and written in standard English?

Reviewer #1: No

Reviewer #2: Yes

5. Review Comments to the Author

Reviewer #1: a) the manuscript can provide further more information regarding the Autism Spectrum Disorder with depressive state and with depression for clarity of the topic and need to do the research. the reviewer wasn't able to capture the clear information regarding the ASD with depressive state and depression. it is understood from science that brain structural differences are seen in persons with ASD and without ASD and this needs to correlated as the study is done with the help of MRI.

b) as far as ASD condition is concerned, the males are more affected by ASD than females. the article should have pronounced sex differences, since females experience twice as much depression as males.

c) As a Rehabilitation Professional- Educationists , Psychologists look for the behavioural output of the subjects. it is not revealed in the study and the reviewer has limitations in further reviewing the manuscript.

Reviewer #2: Title: Connective differences between autism spectrum disorder with depressive state and depression: case-control study

1. How the authors describe the resting state of ASD individuals? Since the people with ASD has different comorbidities and exhibit restlessness in some individuals,

2. Authors pls check for the abbreviations for eg:-DSM 5 is repeated .It can be defined at initial stage itself.

3. Methodology can be given in detail using a block diagram or pictorial representation

4. Technical information is lagging in the manuscript. Such as pre-processed images its metrics and equation related to filtering

5. There is no proof for pre-processing of the functional and structural images and the metrics can be specified.

6. Did the ASD participants were assisted by their caregivers?

7. ASD individuals don’t adjust to social setting. Pls mention the challenges in recording rs fMRI

8. Figure 1 and 2 can be cited with more information.

6. PLOS authors have the option to publish the peer review history of their article (what does this mean?). If published, this will include your full peer review and any attached files.

Reviewer #1: No

Reviewer #2: **Yes: **B. Anandhi

---

## [Author Response · Author response to Decision Letter 0]

5 Mar 2023

Reviewer #1: a) the manuscript can provide further more information regarding the Autism Spectrum Disorder with depressive state and with depression for clarity of the topic and need to do the research. the reviewer wasn't able to capture the clear information regarding the ASD with depressive state and depression. it is understood from science that brain structural differences are seen in persons with ASD and without ASD and this needs to correlated as the study is done with the help of MRI.

Response: We thank the reviewer for this comment. In response, we have performed voxel-based morphometry and added the following sentences to the Methods, Results, and Discussion sections.

Methods

“Structural analysis

Voxel-based Morphometry

We used SPM12 to estimate differences between depression and ASD with depressive state. Preprocessed structural images were smoothed with a 6-mm Gaussian kernel of FWHM to increase the S/N ratio. 

Post-processed structural images were divided into two groups: ASD with depressive state and depression. First, we constructed a design matrix to adopt the general linear model (GLM) and identify regions significantly related to the differences between the two groups. We also performed estimation to validate the independently distributed residuals. Finally, we analyzed the two-sample t-test using the standard parametric procedure to test the hypothesis.[21]”

Results

“Volumetric findings

There was no significant difference between ASD with depressive state and depression.”

Discussion

“We performed voxel-based morphometry using SPM12 to compare the structural differences between ASD and depression.[21] Our results showed no significant structural differences between these two groups, although previous studies demonstrated variable structural abnormality in ASD subjects presenting with increased and decreased volumes in specific areas of the brain.[36-39] ASD subjects in previous studies varied in both age and clinical background (comorbidity, educational and genetic profiles). Compared with healthy controls adult ASD subjects, Ecker et al showed that adult ASD subjects had increased gray matter volume in the anterior temporal and dorsolateral prefrontal regions and decreased volume in the occipital and medial parietal regions, despite there being no significant difference in whole brain volume.[36] In the present study, ASD subjects showed no structural differences compared to subjects with depression. However, subjects with depression were known to display significant volume reductions compared to healthy controls, especially in the hippocampus and amygdala.[40-41] We assumed that the differences in morphological alterations in each study could be due to differences in clinical background. Therefore, our results indicated some structural abnormality in ASD subjects that should be reexamined in healthy controls.”

b) as far as ASD condition is concerned, the males are more affected by ASD than females. the article should have pronounced sex differences, since females experience twice as much depression as males.

Response: We thank the reviewer for this comment. We added following paragraphs in the Discussion section.

“In this study, the numbers of male and female subjects were equal in both the groups. Although there were slightly more male subjects with ASD and slightly fewer male subjects with depression, there was no statistically significant difference in gender distribution. In general, ASD is more than 4 times more common among males than among females, and depression is twice as common among females.[22-24] The equal number of males and females in this study was considered to match that in a previous study because of the opposite gender distribution between ASD and depression. However, the sample size in this study was too small to analyze gender differences. In the case that no gender differences were found in the population presenting with depression, other psychiatric disorders may need to be considered.”

c) As a Rehabilitation Professional-Educationists, Psychologists look for the behavioural output of the subjects. it is not revealed in the study and the reviewer has limitations in further reviewing the manuscript.

Response: We thank the reviewer for this comment. At our institution, patients referred from other institutions are examined, psychologically tested, and diagnosed clinically by an experienced psychiatrist. In this study, patients showed symptoms of depressive state, as shown in the inclusion criteria. However, subsequent physical examination and psychological testing revealed that the depressive state was associated with ASD rather than depression. Many patients received rehabilitation and treatment at the referral institution after the diagnosis was made. It was therefore difficult to consider outputs in more detail. We would appreciate the reviewer’s understanding on this matter. 

Reviewer #2: Title: Connective differences between autism spectrum disorder with depressive state and depression: case-control study

1. How the authors describe the resting state of ASD individuals? Since the people with ASD has different comorbidities and exhibit restlessness in some individuals,

6. Did the ASD participants were assisted by their caregivers?

7. ASD individuals don’t adjust to social setting. Pls mention the challenges in recording rs fMRI

Response: We thank the reviewer for this comment. As pointed out, MRI examinations are considered difficult for patients with ASD, ADHD, and PDD, and ASD was reviewed. All participants in this valuable review were under 20 years of age, and most were children around 10 years of age. In contrast, our target patients were aged over 20 years and were willing to participate in this MRI study. In addition, he presented to the hospital in a depressed state, which is also considered to be the reason why there was no interruption due to movement during the examination and no deterioration of the image. All patients did not require caregiver attendance. However, as the reviewer pointed out, testing is expected to be difficult in general; we would therefore like to state in the Discussion section that the subject in this case was biased toward ASD who showed depressive state. We have revised the relevant paragraphs in Methods and Discussion sections according to these comments.

2. Authors pls check for the abbreviations for eg:-DSM 5 is repeated .It can be defined at initial stage itself.

Response: We thank the reviewer for this comment. We have modified the manuscript accordingly. 

3. Methodology can be given in detail using a block diagram or pictorial representation

4. Technical information is lagging in the manuscript. Such as pre-processed images its metrics and equation related to filtering

5. There is no proof for pre-processing of the functional and structural images and the metrics can be specified.

Response: We thank the reviewer pointing this out. We added details on preprocessing accordingly. In addition, we created a block diagram explaining the processing method conducted in our study. Regarding the metrics, we did not add a corresponding diagram and explanation because they are described on the relevant web page.

8. Figure 1 and 2 can be cited with more information.

Response: We thank the reviewer for this comment. We have inserted the necessary citations in the figures accordingly.

---

## [Decision Letter · Decision Letter 1]

15 May 2023

PONE-D-22-14224R1Connective differences between autism spectrum disorder with depressive state and depression: case-control study.PLOS ONE

Dear Dr. Kaneko,

Thank you for submitting your manuscript to PLOS ONE. After careful consideration, we feel that it has merit but does not fully meet PLOS ONE’s publication criteria as it currently stands. Therefore, we invite you to submit a revised version of the manuscript that addresses the points raised during the review process.

Reviewer #2 raised a number of further issues. Authors are encouraged to evaluate if amending the manuscript according to reviewer suggestions. Please include a rebuttal letter detailing which concerns have been addressed (and how), and which concerns have not been addressed (and why). Note that I'm not implying that all the concerns must be addressed. I'll take a final decision without further reviewers involvement.

We look forward to receiving your revised manuscript.

Kind regards,

Federico Giove, PhD

Academic Editor

PLOS ONE

Journal Requirements:

Reviewers' comments:

Reviewer's Responses to Questions

**Comments to the Author**

1. If the authors have adequately addressed your comments raised in a previous round of review and you feel that this manuscript is now acceptable for publication, you may indicate that here to bypass the “Comments to the Author” section, enter your conflict of interest statement in the “Confidential to Editor” section, and submit your "Accept" recommendation.

Reviewer #2: (No Response)

2. Is the manuscript technically sound, and do the data support the conclusions?

Reviewer #2: Partly

3. Has the statistical analysis been performed appropriately and rigorously? 

Reviewer #2: N/A

4. Have the authors made all data underlying the findings in their manuscript fully available?

Reviewer #2: No

5. Is the manuscript presented in an intelligible fashion and written in standard English?

Reviewer #2: No

6. Review Comments to the Author

Reviewer #2: Dear Authors,

1.Authors pls expalin what is the dfifference between depressive state and depression.? Does the authors trying to find out mild , moderate or severe level of depression?

2.The authors have mentioned the preprocessing methods such as slicing, band pass filtering, smoothing and normalization but there is axial information of the filtered image , sliced image or normalized image ?Probably you can tabulate the results of each step and its S/N ratios achieved.

3.Though you use CONN connectivity tool box , need to mention the actual analysis between depressive state and depression this analysis. Pls give the raw image of Depression and Depressive state image

4 Authors claim that the statistical analysis by performing the F test, has right supramarginal gyrus (SMG) (p-FDR < 0.04*) and decreased connectivity to the left hippocampus (p-FDR < 0.02*) and para-hippocampus (p-FDR < 0.02*). What tool is used for this test ?.What are the hypothesis? Pls mention the Mean , STD from the statistical analysis.

5.Pls check the entire manusript for phraseaology, grammar ,and the connectivity between each section.

6.Improve the discussion section

7. PLOS authors have the option to publish the peer review history of their article (what does this mean?). If published, this will include your full peer review and any attached files.

Reviewer #2: **Yes: **Dr.B.Anandhi

---

## [Editor Report · Decision Letter 2]

12 Jul 2023

PONE-D-22-14224R2Connective differences between patients with depression with and without ASD : a case-control studyPLOS ONE

Dear Dr. Kaneko,

Thank you for submitting your manuscript to PLOS ONE. After careful consideration, we feel that it has merit but does not fully meet PLOS ONE’s publication criteria as it currently stands. Therefore, we invite you to submit a revised version of the manuscript that addresses the points raised during the review process.

The manuscript is fine, it can be accepted from a scientific standpoint. The data access policy stated in the manuscript is not acceptable. Authors must report in the article text the same policy declared in the forms, and in particular name and email of the person to be contacted to get access to the data. Please note that, even after including these details in the manuscript, the policy of your institution is too restrictive and incompatible with open data principles. For the future, if authors plan to submit to journals that require open data, ethical consent and local policies must be amended to allow unconditional sharing of anonymized data.

We look forward to receiving your revised manuscript.

Kind regards,

Federico Giove, PhD

Academic Editor

PLOS ONE
---

## [Author Response · Author response to Decision Letter 2]

22 Jul 2023

Journal Requirements:

1. The manuscript is fine, it can be accepted from a scientific standpoint.

The data access policy stated in the manuscript is not acceptable. Authors must report in the article text the same policy declared in the forms, and in particular name and email of the person to be contacted to get access to the data.

Response: Thank you for reviewing my manuscript.

We have changed our data access policy in the manuscript to be fit in the form.

Response: We have added "doi" to the No. 35 reference. In addition, we changed the style of No. 36 to that of the Book.

There were no comment from reviewer. So, we have submitted a revised version of the paper we submitted on July 9.

---

## [Editor Report · Decision Letter 3]

26 Jul 2023

Connective differences between patients with depression with and without ASD : a case-control study

PONE-D-22-14224R3

Dear Dr. Kaneko,

We’re pleased to inform you that your manuscript has been judged scientifically suitable for publication and will be formally accepted for publication once it meets all outstanding technical requirements.

Kind regards,

Federico Giove, PhD

Academic Editor

PLOS ONE
---

## [Editor Report · Acceptance letter]

3 Aug 2023

PONE-D-22-14224R3 

Connective differences between patients with depression with and without ASD : a case-control study 

Dear Dr. Kaneko:

I'm pleased to inform you that your manuscript has been deemed suitable for publication in PLOS ONE. Congratulations! Your manuscript is now with our production department. 

Kind regards, 

on behalf of

Dr. Federico Giove 

Academic Editor

PLOS ONE